# Duration of Neonatal Antibiotic Exposure in Preterm Infants in Association with Health and Developmental Outcomes in Early Childhood

**DOI:** 10.3390/antibiotics12060967

**Published:** 2023-05-26

**Authors:** Nancy Deianova, Nanne K. de Boer, Hafsa Aoulad Ahajan, Cilla Verbeek, Cornelieke S. H. Aarnoudse-Moens, Aleid G. Leemhuis, Mirjam M. van Weissenbruch, Anton H. van Kaam, Daniel C. Vijbrief, Chris V. Hulzebos, Astrid Giezen, Veerle Cossey, Willem P. de Boode, Wouter J. de Jonge, Marc A. Benninga, Hendrik J. Niemarkt, Tim G. J. de Meij

**Affiliations:** 1Department of Pediatric Gastroenterology, Emma Children’s Hospital, Amsterdam Gastroenterology Endocrinology Metabolism Research Institute, Amsterdam UMC, 1105 AZ Amsterdam, The Netherlandst.demeij@amsterdamumc.nl (T.G.J.d.M.); 2Department of Pediatric Gastroenterology, Amsterdam UMC Location University of Amsterdam, Amsterdam Reproduction & Development Research Institute, 1105 AZ Amsterdam, The Netherlands; 3Department of Neonatology, Máxima Medical Center, 5504 DB Veldhoven, The Netherlands; hendrik.niemarkt@mmc.nl; 4Department of Gastroenterology and Hepatology, Amsterdam Gastroenterology Endocrinology Metabolism, Amsterdam University Medical Centre, Vrije Universiteit Amsterdam, 1105 AZ Amsterdam, The Netherlands; 5Department of Neonatology, Emma Children’s Hospital, Amsterdam Reproduction and Development Research Institute, 1105 AZ Amsterdam, The Netherlands; 6Department of Neonatology, University Medical Center Utrecht, Wilhelmina Children’s Hospital, 3584 CX Utrecht, The Netherlands; 7Department of Neonatology, Beatrix Children’s Hospital, University Medical Center Groningen, 9713 GZ Groningen, The Netherlands; 8Department of Neonatology, Isala Hospital, Amalia Children’s Center, 8025 AB Zwolle, The Netherlands; 9Department of Neonatology, University Hospitals Leuven, 3000 Leuven, Belgium; 10Department of Neonatology, Radboud University Medical Center, Radboud Institute for Health Sciences, Amalia Children’s Hospital, 6525 XZ Nijmegen, The Netherlands; 11Tytgat Institute for Liver and Intestinal Research, Amsterdam Gastroenterology Endocrinology Metabolism Research Institute, Amsterdam UMC, University of Amsterdam, 1105 AZ Amsterdam, The Netherlands; w.j.dejonge@amsterdamumc.nl

**Keywords:** neonatal antibiotic exposure, preterm, psychomotor development, atopy, pediatric constipation, growth

## Abstract

Over 90% of preterm neonates are, often empirically, exposed to antibiotics as a potentially life-saving measure against sepsis. Long-term outcome in association with antibiotic exposure (NABE) has insufficiently been studied after preterm birth. We investigated the association of NABE-duration with early-childhood developmental and health outcomes in preterm-born children and additionally assessed the impact of GA on outcomes. Preterm children (GA < 30 weeks) participating in a multicenter cohort study were approached for follow-up. General expert-reviewed health questionnaires on respiratory, atopic and gastrointestinal symptoms were sent to parents of children > 24 months’ corrected age (CA). Growth and developmental assessments (Bayley Scales of Infant and Toddler Development (BSID) III) were part of standard care assessment at 24 months’ CA. Uni- and multivariate regressions were performed with NABE (per 5 days) and GA (per week) as independent variables. Odds ratios (OR) for health outcomes were adjusted (aOR) for confounders, where appropriate. Of 1079 infants whose parents were approached, 347 (32%) responded at a mean age of 4.6 years (SD 0.9). In children with NABE (97%), NABE duration decreased by 1.6 days (*p* < 0.001) per week of gestation. Below-average gross-motor development (BSID-III gross-motor score < 8) was associated with duration of NABE (aOR = 1.28; *p* = 0.04). The aOR for constipation was 0.81 (*p* = 0.04) per gestational week. Growth was inversely correlated with GA. Respiratory and atopic symptoms were not associated with NABE, nor GA. We observed that prolonged NABE after preterm birth was associated with below-average gross-motor development at 24 months’ CA, while a low GA was associated with lower weight and stature Z-scores and higher odds for constipation.

## 1. Introduction

Approximately 90% of very-low-birth-weight infants are, often empirically, exposed to antibiotics within the first weeks of life [1,2,3]. Since their discovery by Sir. A. Fleming (1881–1955), antibiotics have had major impact on health and increased life expectancy significantly [4]. Indeed, antibiotics are indispensable in neonatal care [5]. However, the success of and the reliance on antibiotics also has unwanted side effects, and inappropriate use of antibiotics has been related to adverse short-term outcomes and antibiotic resistance [4,6,7,8].

There is an additional increasing concern because of putative long-term adverse health effects correlated to antibiotics [9,10]. A longer duration of neonatal antibiotic exposure (NABE) might have an effect on microbiota diversity, which has been proposed to lead to impaired immune defense mechanisms with potential adverse effects in the long term [6,9,10,11,12]. Currently, antibiotic stewardship guidelines for suspected sepsis discourage antibiotic continuation upon negative blood culture results from 48 to 72 h to minimize potential adverse effects of antibiotics [13,14]. To optimize these guidelines, continuous research on potential effects of antibiotics is needed.

In preterm infants, however, the low sensitivity of blood cultures and fear of the consequences of missed infection leads to prolongation of the antibiotics course [14,15]. Prolonged antibiotic exposure in the first days of life has been associated with life-threatening diseases such as necrotizing enterocolitis (NEC) and late-onset sepsis (LOS) in the first weeks after preterm birth [8]. In the long term, antibiotics in term-born infants have been associated with an increased risk of asthma, atopy, obesity and autism spectrum disorders in childhood [16,17,18].

Based on in vivo observations in animals and humans, it is currently hypothesized that pre- and postnatal exposure to antibiotics might interfere with organ and immune system development through several mechanisms [19,20]. Antibiotics can alter intestinal microbiota composition and delay or change early colonization, creating a less hospitable environment for bacteria that produce beneficial short-chain fatty acids (SCFA), e.g., *Bifidobacteria* and *Bacteroides* species [21,22,23]. This may impair priming of the immune system, facilitating chronic inflammation and potentially interfering with normal growth and respiratory, gastrointestinal and psychomotor development [24,25,26].

In preterm infants, the impact of early-life antibiotic exposure on health outcome beyond the neonatal period is largely unclear, despite the high rates of prolonged NABE. In the current study, we aimed to investigate the association of duration of NABE after preterm birth (gestational age (GA) < 30 weeks) with growth and neurodevelopmental impairments, wheezing, atopy and constipation. GA, a known risk factor for adverse outcomes, was included as a secondary potential risk factor in the analysis of our cohort. 

## 2. Methods

### 2.1. Design and Subjects

This follow-up study was embedded in an ongoing longitudinal multicenter cohort study including preterm neonates since 2014, with the primary aim to identify fecal biomarkers for neonatal NEC and late-onset sepsis (LOS) [27]. In 2020, follow-up of this cohort was initiated to investigate the association between neonatal microbiota and microbiota-altering factors and health outcomes beyond the neonatal period. During the neonatal period (the first 28 days of life), clinical variables were collected prospectively from birth to the age of 28 days or until discharge, whichever came first.

Follow-up was conducted cross-sectionally between June 2020 and May 2022 in surviving participants. For this purpose, a questionnaire for parents was designed based on current literature (Appendix A) and evaluated by local experts in the field of pediatric pneumology, allergology, gastroenterology and neonatology to assess childhood health outcome. Questionnaires were designed in Dutch and translated to French and English.

As depicted in Figure 1, the study consisted of a one-time parent’s reported, written questionnaire-based health assessment beyond 24 months’ corrected age (CA) of children born between October 2014 and July 2019. Additionally, retrospective data on weight and height, as well as Bayley Scale of Infant and Toddler Development, 3rd edition (BSID-III) scores, were collected [28].

The study was conducted in eight neonatal intensive care centers: Amsterdam University Medical Centers (two locations), Maxima Medical Center, Isala Zwolle, University Hospital Leuven, University Medical Center Groningen, University Medical Center Utrecht and Radboud University Medical Center.

If parents did not respond to contact by telephone, they were approached by paper mail. Exclusion criteria at initial time of enrollment were congenital gastrointestinal anomalies, including intestinal atresia and Hirschprung’s disease, and Down syndrome. Additional exclusion criteria for the follow-up period were insufficient verbal and/or written language knowledge of Dutch, English and French. In cases of non-response, parents were reminded telephonically. 

Each participant of the neonatal cohort was appointed a study-ID to ensure de-identification. Data provided by the parents were stored on an encrypted data capture tool (Castor^®^ Electronic Data Case 22, Amsterdam, The Netherlands). Ethical approval was granted by the local Medical Ethical Committee (registration number: A2020.219).

### 2.2. Data Collection

Perinatal and neonatal data, as well as BSID-III scores and weight and height at 24 months’ CA, were collected from the electronic patient record (EPR). BSID-III, as well as GA-normalized weight and stature Z-scores, were administrated during the regular visit for follow-up of preterm-born infants at 24 months’ CA. BSID-III was assessed by a team of developmental psychologist and physiotherapists at every participating center as part of a national neonatal follow-up program.

Duration of NABE (with precision of 1 day) was calculated from the pre-existing database created during the subjects’ neonatal period. Other perinatal characteristics included gestational age, mode of delivery, mother’s parity, multiple births, gender, Apgar scores, birth weight, birth weight Z-score, neonatal enteral nutrition type and development of NEC (defined as stage ≥ IIA according to the modified Bell’s stage criteria by Kliegman and Walsch), culture-proven LOS and culture-proven meningitis.

Childhood outcomes, other than BSID-III and weight and height, were derived from the questionnaire completed by the parents after 24 months’ CA. Questionnaire items on symptoms of bronchial constriction, rhinitis and atopic dermatitis were based on adapted questions from the International Study of Asthma and Allergies in Childhood (ISAAC) and from a large Dutch cohort study on respiratory health (Appendix A) [29,30]. Survey items for constipation were based on the Rome IV criteria for functional constipation, as described in Appendix A [31]. Based on previous literature, we expected 95% of the pediatric constipation to be functional and therefore applied the ROME IV criteria, without formally investigating organic causes of pediatric constipation [32]. Hirschprung’s disease was the only potential organic cause of constipation that was a formal exclusion criterion.

BSID-III Cognitive and Motor scores were classified as below-average below 90, as described in the Dutch BSID-III scoring manual and recently supported by Månsson et al. [28,33]. The cut-off for below-average gross and fine motor scores was 8, in accordance with the aforementioned manual [28]. When development according to BSID-III assessment was reported as developmental age, rather than a numeric score, the conclusion of the assessment, also based on the scoring manual, was used for determination of below-average vs. (above-)average scores.

### 2.3. Statistical Analysis

Statistical analyses were conducted using the Statistical Package for Social Sciences (SPSS) version 26.0 (IBM, Armonk, NY, USA). First, baseline perinatal and neonatal characteristics were compared between a representative sample of children included in the original neonatal cohort study and those participating in the current follow-up study. Secondly, baseline clinical and demographic characteristics were depicted as number and percentage for discrete variables and as mean/median and standard deviation (SD)/interquartile range (IQR) for continuous variables.

The association of the duration of NABE (continuous variable) with the main outcomes was assessed by uni- and multivariate linear and logistic regression. All regression analyses were first performed uncorrected and then corrected for relevant confounding. Confounders were defined as variables significantly associated with both the independent and dependent variables (Pearson correlation *p*-value < 0.05). Variables that were tested as confounders are depicted in a direct acyclic graph (Appendix A); gestational age [34], birth weight percentile [35], parental education [36], invasive neonatal respiratory failure [37], sex, mode of delivery, length of hospital stay at the neonatal intensive care unit, neonatal NEC/sepsis/meningitis, age at questionnaire and formula feeding during infancy [38] were tested for each outcome variable. Only univariate analysis was performed, in case no variables were correlated to both independent and dependent variables.

Results from the linear regression analyses were reported as regression coefficient with the respective 95% confidence interval (95% CI). Results from the logistic regression were reported as odds ratios (OR) and adjusted OR (aOR), along with the respective 95% CI. As a secondary objective, GA (per 7 days) was analyzed as a risk factor for poor childhood health outcome analogically to NABE. 

## 3. Results

Of 1190 infants (GA at birth < 30 weeks), born between October 2014 and July 2019 in the eight participating centers, parents of 1079 children were approached. Completed questionnaires and informed consent forms were received from parents of 347 (32%) children; the children had a median CA of 53 months (Figure 2), were more often singleton and had less sepsis and NEC. Moreover, GA (+1 day) and birth weight (+42 g) were higher compared to non-responders (Table 1). 

Baseline characteristics of the group of responders are depicted in Table 2. Mean (SD) duration of NABE was 9 days (SD = 7). One-hundred thirty-two (38%) of the included preterm infants were exposed to neonatal antibiotics for at least two weeks, of whom 81 had neonatal sepsis, meningitis and/or NEC. For every extra week of gestation, NABE decreased with a mean of 1.6 days (*p* < 0.001).

### 3.1. Growth and Neurodevelopment at 24 Months’ Corrected Age

Mean weight and stature at 24 months’ CA were 12 kg and 87 cm, respectively (Table 2). Weight and stature Z-scores at 24 months’ CA, according to GA-normalized growth charts, were not associated with NABE (Table 3). In contrast, every extra 7 days of GA at birth was associated with a 0.10 higher Z-score for weight and 0.08 higher corrected Z-score for stature at 24 months’ CA.

Median BSID-III cognitive and motor scores were within normal range (101 and 102, respectively) [28]. BSID cognitive, composite motor and fine motor scores were not associated with NABE. The association of a below-average gross motor developmental score (<8) with NABE remained significant after correction for gestational age; OR for every five days of additional NABE was 1.28 [1.06–1.54] (*p* = 0.04) (Table 3). An increased incidence of below-average motor scores (*p* = 0.03) was seen with decreasing gestational age at birth, but not after correction for confounding (*p* > 0.10), as depicted in Appendix A. 

### 3.2. Gastrointestinal Symptoms at Time of Survey

Forty-seven (14%) children were reported to meet at least two ROME IV criteria for functional constipation or being treated for constipation. There was no association between NABE and pediatric constipation. Odds for constipation decreased with increasing gestational age (OR 0.81 per week of gestation, *p* = 0.04), independent from confounding (Appendix A).

### 3.3. Respiratory Symptoms in Early Childhood

Parents of 114 children (35%) reported to have ever sought medical attention (general practitioner or emergency room) for their child due to acute lower respiratory tract symptoms. Sixty-five children (19%) were reported to have ever been hospitalized due to lower respiratory tract disease (Table 3). Thirty-six (11%) children were affected by moderate-to-severe lower respiratory symptoms at the time of survey, needing at least weekly inhalation therapy over the past month. About one in five subjects had undergone surgery for recurrent upper respiratory infections (tympanostomy tubes, nasal polypectomy and/or tonsillectomy). None of the respiratory outcomes were associated with NABE, nor GA (Table 3 and Appendix A, respectively). 

### 3.4. Atopic Symptoms in Early Childhood

Fifty-five (17%) children were suspected of food allergies by their parents, of which nine (3%) were reported to be confirmed by a health care professional. Forty-three (14%) children were reported to have had symptoms of allergic rhinoconjunctivitis over the past year, and thirty-five (10%) were reported to have ever had atopic dermatitis. None of the respiratory and skin symptoms were correlated with NABE, nor GA (Table 3, Appendix A).

## 4. Discussion

In this multicenter cohort study in infants born preterm (<30 weeks GA), we aimed at investigating the association of NABE with developmental and health outcomes after 24 months of corrected age. Almost all infants (97%) were exposed to antibiotics, with 38% being exposed for at least 14 days within the first 28 days of life. The duration of NABE was found to be associated with below-average BSID-III gross motor scores (<8) at 24 months’ CA. In line with our results, motor skills are shown to be impaired in adult mice after administration of antibiotics [39,40]. It is hypothesized that the gut microbiota, which is affected by antibiotics, plays a role in cognitive and neuromotor functions. The proposed mechanism of the negative effect of NABE on development is by impairing cell signaling of the microbiota via neural (vagus nerve), humoral (cytokines, SCFA and long-chain fatty acids), endocrine and immune modulators [23,41,42]. In mice, the decreased motor performance after antibiotic exposure could be alleviated by restoring the gut microbial composition with probiotics administration [40]. 

NABE has, furthermore, been suggested to affect adult cognitive development in term-born mice [43]. In contrast to our results, this has been supported by previous observations, showing worse school performance in children with a history of NABE [44]. The authors of that study, however, do not mention early-life infection as a potential confounding factor [44]. The contribution of infection in school performance should be taken into account as it has previously been associated with poor neurodevelopmental outcome [45]. Furthermore, a small study in 24 very-low-birth-weight preterm infants showed differences in microbiota composition in the first weeks of life in children with early childhood cognitive impairment [46].

Pediatric constipation, respiratory and atopic symptoms were not associated with NABE. In contrast, in term-born children, chronic lower respiratory symptoms due to asthma and other atopic conditions have been correlated to early-life antibiotic exposure [17,47]. This discrepancy with the current preterm cohort might be due to a potentially different mechanism of atopy development after preterm birth, partly illustrated by differences in the atopic march [48]. Similarly, gestational age was not associated with wheezing, nor atopic conditions, as opposed to previous data on wheezing being inversely correlated and atopic conditions being positively correlated with GA [44,46,47]. These studies, however, compare preterm- with term-born children and thus include more heterogeneous groups of children [44,46,47]. In the current, rather homogenous group of children born between 24 and 30 weeks of gestation, the effect of GA might be smaller and more challenging to detect.

Anthropometric parameters were only associated with GA and not NABE. Lower gestational age was associated with lower GA-normalized Z-scores for weight at 2 years’ CA, in line with a recent report comparing preterm- vs. term-born children [49,50]. Interestingly, constipation after 24 months’ CA was additionally associated with GA. Based on previous literature, we expected the vast majority (95%) of the children with constipation to be functional [32]. The potential underlying mechanism of the association between GA and functional constipation is to be explored. In adults, severe dysmotility is associated with underlying vagus nerve dysfunction due to autonomic dysfunction [46]. Whether autonomic dysfunction, as seen in preterm newborns, can contribute to functional constipation in childhood remains to be elucidated [47].

Future studies integrating clinical and early-life microbiota data are warranted to validate our results and test the hypothesized mechanisms of action of NABE and GA on health. Large observational studies might identify potential microbial signatures associated with disease. This could lead to new basic research opportunities and the design of preventative and therapeutic strategies. 

### Strengths and Limitations

This study has several strengths. It contributes to mapping health problems after preterm birth in relation to early antibiotic exposure and GA, assessing relatively large numbers of preterm born infants. We followed preterm children longitudinally, and detailed information on daily base was collected on antibiotic use and indication of antibiotic administration in the first month of life. Growth and developmental scores were assessed by trained professionals. The survey was based on existing validated questionnaires, including the ISCAAC questionnaires [29]. In addition, every section of the questionnaire provided free text space, enabling additional coding. Additionally, statistical analysis was performed systematically and was designed to avoid overfitting.

There are, however, several limitations to this study. First, inherent to the observational methodology, no causal role of NABE and GA on health could be tested. Secondly, the age at which the questionnaire was completed varied between two and seven years, which could be a confounding factor for airway problems and atopy. The influence of age at which the questionnaire was completed was, however, estimated to be minimal as it did not fulfil the criteria of a confounding factor described in the method section; i.e., it was not correlated with both NABE and health outcomes. A second factor potentially creating bias is the lack of information on prenatal antibiotic exposure and maternal health, which might influence the risk of motor impairment and atopic disease [20,51]. Additionally, as we were not able to assess the type of antibiotics administered, we could not assess whether certain antibiotics groups are associated with different outcomes. Additionally, inherently to the retrospective character of the developmental data collection (BSID-III scores), we were limited by dichotomized reports of the BSID-III scores in some children. Therefore, the number of children analyzed with continuous composite cognitive and motor scores is markedly lower. It remains to be elucidated whether increasing the samples size would reveal an association of developmental scores with NABE.

Another limitation is the relatively low response rate of 32%. This might be because at the time of inclusion at neonatal age, parents were not informed about the follow-up of this study. As it was decided upon follow-up of children after 5 years of inclusions, this part of the study could not be integrated in the standardized visits at the ambulatory and was performed outside the hospital setting [52]. The low response rate might compromise the generalizability of the results, as the participants of the follow-up had lower rates of LOS (27 vs. 32%) and NEC (4% vs. 8%) during the neonatal period, compared to the initial neonatal cohort. They were potentially different in (non-analyzed) demographics, such as neonatal disease severity scores (e.g., Score for Neonatal Acute Physiology with Perinatal Extension-II) and parents’ knowledge about adverse effects of antibiotics [53]. Finally, the definition of the investigated diseases can be subject to bias. Although the survey was based on validated questionnaires, wheezing and allergy questions are based on subjective experiences of parents and are an overestimation of clinical diseases. 

## 5. Conclusions

Our results suggest that duration of NABE and GA might be associated with health complaints in early childhood after preterm birth (GA < 30 weeks). In the investigated cohort, NABE was associated with impaired gross motor development in toddlers, while a low GA was associated with poor weight gain and pediatric constipation. Further research is needed to evaluate the complex interplay between antibiotics, microbiota colonization and health outcome beyond the neonatal age.

## Figures and Tables

**Figure 1 antibiotics-12-00967-f001:**
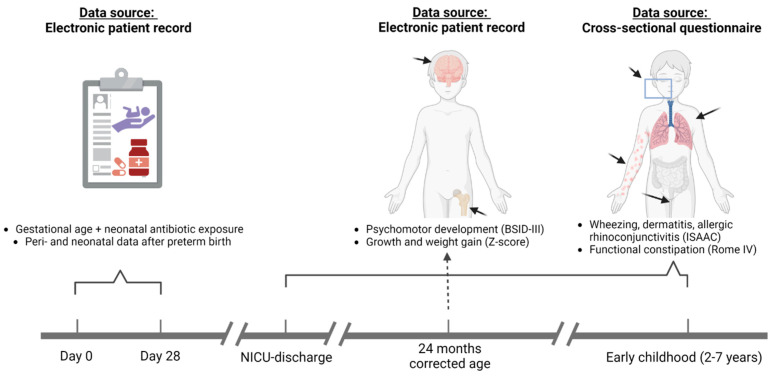
Workflow of the preterm cohort study. During the neonatal period (first 28 days of life), daily clinical variables were registered in neonates born before 30 weeks’ gestation. Follow-up consisted of anthropometry and BSID-III developmental scores at 24 months’ corrected age (CA) and parents’ reported, questionnaire-based health assessment beyond 24 months’ CA. BSID-III, Bayley’s Scale of Infant and Toddler Development—3rd version; ISAAC, International Study of Asthma and Allergies in Childhood; NICU, neonatal intensive care unit; Rome-IV, Rome IV Diagnostic Criteria for Functional Constipation. Created with biorender.com.

**Figure 2 antibiotics-12-00967-f002:**
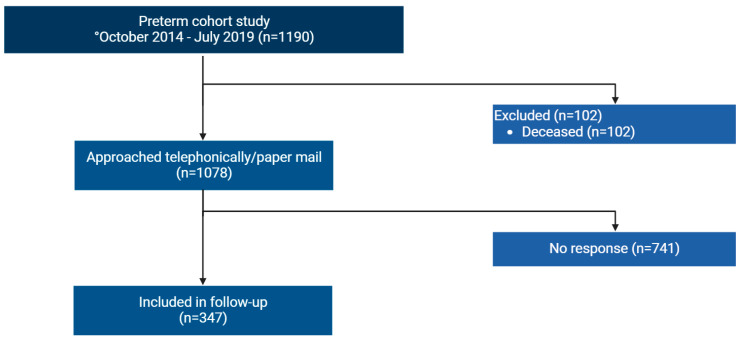
Flowchart of inclusion. Between June 2020 and May 2022, parents of children born in eight neonatal intensive care units that participated in the follow-up study were approached. In total, 347 children were included in the follow-up study.

**Table 1 antibiotics-12-00967-t001:** Comparison of baseline characteristics of initial neonatal cohort and follow-up cohort.

	Initial Cohort (n = 1190)	Follow-Up Cohort (n = 347)	*p*-Value
Gestational age, weeks + days, mean (SD)	27 + 4 (1 + 4)	27 + 5 (1 + 4))	0.03 *
Gender, female, %	47%	45%	0.67
Delivery mode, cesarean, %	52%	48%	0.23
APGAR score 1 min, median [IQR])	6 [4,5,6,7]	6 [3,4,5,6,7]	0.93
APGAR score 5 min, median [IQR])	8 [6,7,8,9]	8 [6,7,8,9]	0.50
Multiplicity (twin or triplet), %	32%	27%	0.02 *
Birth weight, gram, mean (SD)	1025 (271)	1067 (268)	0.01 *
Birthweight, Z-score, mean (SD)	0.04 (1.01)	0.13 (0.99)	0.16
Late-onset sepsis in first 28 days of life, %	34%	27%	0.02 *
Necrotizing enterocolitis, %	8%	4%	0.01 *

* *p*-value < 0.05; IQR, interquartile range; SD, standard deviation.

**Table 2 antibiotics-12-00967-t002:** Baseline demographic and clinical characteristics of all subjects.

	Mean (SD) or n (%)	Missing Data (N)
Infancy		
Duration of neonatal antibiotic exposure (NABE) in the center of birth, days, mean (SD)	9 (7)	0
No NABE, n (%) NABE for max. 4 days, n (%) NABE 5–13 days, n (%) NABE ≥ 14 days (of which with confirmed disease), n (%)	10 (3) 102 (29) 103 (30) 38 (23)	
First gravidity, n (%)	186 (54)	0
First parity, n (%)	228 (66)	0
Invasive ventilation first 28 d of life, n (%)	172 (50)	0
Intraventricular hemorrhage/periventricular leukomalacia grade III-IV, n (%)	21 (6)	1
>20% formula milk in first 28 d, n (%)	61 (21)	55
Neonatal necrotizing enterocolitis, sepsis or meningitis, n (%) Of which non-CoNS sepsis ^b^, NEC, meningitis, n (%)	96 (28) 35 (10)	0
Exposure to mother’s milk ≥ 6 months	18 (5)	8
Moderate to severe bronchopulmonary dysplasia, n (%)	81 (24)	7
Palivizumab administration, n (%)	261 (88)	49
Childhood		
Age at questionnaire, years, mean (SD)	4.6 (0.9)	0
Weight at ca. 24 months’ CA, mean (SD)	11.9 (1.6)	45
Stature at ca. 24 months’ CA, mean (SD)	86.7 (4.6)	50
Ever received medical feeding at home (oral or tube), n (%)	18 (5)	8
Weekly fastfood or ready-made food, n (%)	47 (14)	18
Family history		
Maternal education, university level, n (%)	182 (55)	18
Ethnicity mother, non-Dutch/Belgian, n (%)	62 (19)	16
Ethnicity father, non-Dutch/Belgian, n (%)	51 (15)	14
Both parents’ mother tongue other than Dutch, n (%)	34 (10)	13
At least monthly exposure to nicotine smoke, n (%)	12 (5)	96
Atopic disease parents or siblings, n (%)	150 (44)	7

^b^ sepsis caused by other pathogens than coagulase negative Staphylococci. CA corrected age; N, total number of missing subjects per specific variable; n (%), number and percentage of subtotal number of participants per item; NABE, neonatal antibiotic exposure; SD, standard deviation.

**Table 3 antibiotics-12-00967-t003:** Regression coefficient (continuous variables) and odds ratio (categorical variables) for health per 5 days of neonatal antibiotic exposure, including confounding factors that were used in the multivariate regression analysis.

		Neonatal Antibiotic Exposure (per 5 Days)			
	Mean (SD) or n (%)	Odds Ratio or Regression Coefficient [95%CI]	*p*-Value	Confounders in the Multivariate Regression Analysis	Missing Data (N)
Growth and development					
Weight Z-score at ca. 24 months’ CA, mean (SD)	−0.47 (1.06)	−0.11 [−0.20–−0.02] Adj: 0.08 [−0.17–0.02]	0.01 * 0.10	Gestational age	69
Stature Z-score at ca. 24 months’ CA, mean (SD)	0.14 (1.01)	0.003 [−0.08–0.09] Adj: NA	0.22 NA	/	71
BSID-III Cognitive or Motor score < 90	78 (28)	1.23 [1.02–1.48] Adj: 1.88 [1.05–3.35]	0.03 * 0.23	Invasive ventilation (<28 days’ age)	67
BSID-III cognitive score, mean (SD)	101 (13)	−0.40 [−1.61–0.80] Adj:NA	0.51 NA	/	84
BSID-III cognitive score < 90, n (%)	52 (17)	1.00 [0.80–1.24] Adj: 0.90 [0.71–1.15]	0.99 0.40	/	42
BSID-III overall motor score, mean (SD)	102 (14)	−1.31 [−2.68–0.07] Adj: 0.28 [−0.04–0.60]	0.06 0.09	Time until discharge from neonatal intensive care unit	134
BSID-III overall motor score < 90, n (%)	63 (22)	1.32 [1.09–1.61] Adj: 1.22 [0.98–1.52]	0.005 * 0.08	Gestational age, invasive ventilation (<28 days’ age)	58
BSID-III fine motor score < 8, n (%)	37 (12)	1.16 [0.91–1.47] Adj: 1.09 [0.84–1.41]	0.24 0.52	Gestational age	47
BSID-III gross motor score < 8, n (%)	82 (31)	1.28 [1.06–1.54] Adj: 1.21 [1.00–1.46]	0.01 * 0.04 *	Gestational age	71
Gastrointestinal symptoms					
Pediatric constipation, n (%)	47 (14)	1.08 [0.87–1.33] Adj: 1.06 [0.84–1.34]	0.48 0.61	Invasive ventilation (<28 days’ age)	18
Use of antacid medication, n (%)	13 (4)	1.14 [0.78–1.67] Adj: NA	0.50 NA	/	7
Respiratory symptoms					
Sought medical attention due to LRT ^a^, n (%)	114 (34)	1.12 [0.96–1.32] Adj: 0.94 [0.78–1.13]	0.15 0.52	Gestational age, invasive ventilation (<28 days’ age)	8
Hospitalization due to acute LRT ^a^ diseasec, n (%)	64 (19)	1.01 [0.83–1.23] Adj: NA	0.92 NA	/	7
Ear-nose-throat surgery, n (%)	65 (19)	1.11 [0.91–1.34] Adj: NA	0.30 NA	/	7
Wheezing episode in the past 12 months, n (%)	88 (26)	1.08 [0.91–1.29] Adj: 0.93 [0.76–1.13]	0.38 0.46	Invasive ventilation (<28 days’ age)	10
Bronchodilation past month, n (%)	73 (22)	1.13 [0.94–1.36] 1.00 [0.81–1.23]	0.21 0.99	Invasive ventilation (<28 days’ age)	10
Bronchodilatation (weekly), n (%)	36 (11)	1.17 [0.92–1.48] Adj: 1.02 [0.78–1.34]	0.20 0.88	Invasive ventilation (<28 days’ age)	15
Atopic symptoms					
Suspected food allergies, n (%)	55 (17)	1.18 [0.96–1.44] Adj: NA	0.12 NA	/	10
Medically confirmed food allergies, n (%)	9 (3)	0.69 [0.36–1.31] Adj: NA	0.25 NA	/	11
Atopic dermatitis (ever), n (%)	35 (10)	0.87 [0.66–1.16] Adj: NA	0.35 NA	/	7
Allergic rhinitis or conjunctivitis (past 12 months), n (%)	43 (14)	1.03 [0.81–1.30] Adj: NA	0.82 NA	/	29
Anti-allergic treatment (current), n (%)	6 (2)	1.60 [0.98–2.63] Adj: NA	0.06 NA	/	7

* *p*-value < 0.05 (bold = adjusted *p*-value < 0.05); ^a^ LRT, Lower respiratory tract disease: bronchiolitis, bronchitis, pneumonia, and asthma-like attack; Adj, adjusted regression result based on confounding as defined in the method section; BSID-III, Bayley Scales of Infant and Toddler Development-III-Dutch version; CA, corrected age; CI, confidence interval; n (%), number and percentage of total number; N number of subjects with missing data for each particular variable; NA, not applicable; SD, standard deviation.

## Data Availability

All data used for this study are available upon reasonable request.

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
