# Peer review of "Duration of Neonatal Antibiotic Exposure in Preterm Infants in Association with Health and Developmental Outcomes in Early Childhood"

_antibiotics, 2023, doi:10.3390/antibiotics12060967_

Round 1
Reviewer 1 Report
I suggest the inclusion of other references to complement the information contained in lines 53 to 55 to correlate immune defense with microbiome and resistome. I suggest reading “Reyman M, van Houten MA, Watson RL, Chu MLJN, Arp K, de Waal WJ, Schiering I, Plötz FB, Willems RJL, van Schaik W, Sanders EAM, Bogaert D. Effects of early-life antibiotics on the developing infant gut microbiome and resistome: a randomized trial. Nat Commun. 2022 Feb 16;13(1):893. doi: 10.1038/s41467-022-28525-z. PMID: 35173154; PMCID: PMC8850541.”
Citation nº 8 (lines 57 to 61) needs to be updated and supplemented with information. I suggest reading and evaluating the article “Rallis D, Giapros V, Serbis A, Kosmeri C, Baltogianni M. Fighting Antimicrobial Resistance in Neonatal Intensive Care Units: Rational Use of Antibiotics in Neonatal Sepsis. Antibiotics (Basel). 2023 Mar 3;12(3):508. doi: 10.3390/antibiotics12030508. PMID: 36978375; PMCID: PMC10044400.”
In the conclusions, the issues of functional constipation in the conclusions were confused, since they do not report the organic causes and in lines 198 and 199 the authors state that “There was no association between NABE and functional constipation, which is repeated in lines 21 and 22 of page 11 and do not reflect the Title of the article.
Author Response
Please refer to the comments of the attached file.

Reviewer 2 Report
This is an interesting cohort study of prematurely born infants and health outcomes at 24 months corrected age (neurocognitive development) or 2-7 years (atopy & constipation). Similar studies have been published previously, including 2 referenced in the discussion. The topic is of great interest as it has therapeutic and developmental implications. I am not a statistician but I believe appropriate methods were employed. Nor am I a pediatrician or neonatologist , so as a reviewer may be out of my element but have endeavored to understand the study and provide constructive criticism.
Major concerns: The investigators examined neonatal antibiotic exposure (NABE) and found a (adjusted) correlation between NABE and % with gross motor below a score of 9. Overall motor score (which I assume includes gross motor score items) was not significantly different, nor was % overall motor score below a score of 90. I am concerned that the amount of statistical scrutiny (over analysis) in the data analysis opens the door for finding by chance or spurious finding associated with severity of illness leading to use of antibiotics.
The authors are careful in language to state an association was found and not claim the relationship is causal but discuss mechanisms where by it is conceivably causal in the discussion. Unfortunately, in this age of disinformation tragedies have resulted from impressionable parents refusing care of acute infection (eg. https://www.cbc.ca/news/canada/calgary/david-collet-stephan-ezekiel-trial-decision-1.5288343). There is a possibility that this study (& other studies) could be read by concerned parents, misinterpreted and similar tragedies could occur. I believe there can be greater emphasis that acute infection, for which antibiotics are absolutely life-saving in very fragile premature infant with sepsis. Since such misintrepretation is often based on the title, I would suggest that is could be worded less suggestively eg. "Cohort study of premature infants found an association between antibiotic duration and one neurocognitive developmental outcome" or simply "Results of a cohort study of premature infants and perinatal events on outcomes neurocognition, function constipation and atopy" . If abstract word counts permits, some statement recognizing the life-saving role that antibiotics have in neonatal care.
Line by line comments for consideration of editor/authors follow; I will also attach my working copy for easy referral.
Title - please see above comments.
Line 39 (abstract) - only 32% response at 4.6 years. Are responders more likely to be parents of children with poor outcomes or notions of antibiotics as harmful?
Line 41 (abstract) - Does not agree with Table 3 - Gross Motor BSID <9 ? Overall motor score <90 - Adj OR: 1.22 [0.98 – 1.52]
Line 76 (intro) - Close association of GA and NABE - yes seems like statistically challenging. And are two children with the same GA similar otherwise? It would seem a lot of other factors, even infection, could impact longterm outcomes.
Line 85 (methods) - Maybe put citation [20] in first sentence of paragraph?
Line 118 (data collection)
Line 123 (data collection) - Explain this process in greater detail here. When were outcomes selected? Before the questionaire was developed? After the data from the questionaire? Who are the experts and how was validity of the outcomes and questions established.
Line 137 (data collection) - Is there validity for fine and gross motor score<9 from this or other research as well? Please be specific and include additional citation if necessary. Otherwise how was fine and gross motor threshold of 9 established?
Line 190 (results) - same issue as in abstract - Table 3 shows Gross Motor Score <9 adjusted OR to be significant +1.21 (1.00-1.46) . Odds ratios cannot be negative numbers is this a log OR?
Table 2 - Descriptives - What does this column mean?
Table 3 - Second column ('N') - Not clear what these numbers are to me.Was this not based on respondents = 347? THird column "Mean(SD)/n%" -please put "or" rather than "/" , its very confusing. Correct negatives OR or report as Log OR.
Line 16 - Discussion - Presumably, NABE were given due to infection of some kind. How can one determine the role of infection in school performance. Did this study [37] adjust for severity of infection?
Line 44 - Discussion - Should be acknowledged that when one makes statistical associations of a bunch of exposure variables with a bunch of outcome variables that a p value <0.05 may come up by chance. In adjusted analyses gross motor score less than 9 (dichotomized variable) had p exactly on the threshold but not overall motor score (continuous variable) . But both flirting with level of significance. Is there rationale for using <9 as the cutoff for gross motor score ? If 8 or 10 had been used would this have the same result? In general, dichotomizing continuous variables is discouraged for multiple reasons. (https://www.bmj.com/content/332/7549/1080.1) So I would consider this as hypothesis generating and should be interpreted with caution and further research is needed for firm conclusions. It may infact be an encouraging that there is not a profound signal of NABE to overall motor score.

Author Response

(The authors gave the same response as above.)

Reviewer 3 Report
it would be more useful to also divide by age range: 2-4, 5-7, and also specify the type of antibiotic used
Author Response

(The authors gave the same response as above.)

Round 2
Reviewer 2 Report
Appreciate the changes in the title , abstract and introduction and change in tone, appropriate with the findings of this cohort study. I'm also appreciative of the author's time spent explaining their use methods and use of scales as they did.
The limitations have been elaborated upon and clearly stated. I feel this is more tempered and conservative reporting.
Overall I think this manuscript is much improved and should be published. I did not find any further need for amendments.